# Towards Robustness Against Natural Language Word Substitutions

**Xinshuai Dong**
Nanyang Technological University, Singapore
dongxinshuai@outlook.com

**Anh Tuan Luu**
Nanyang Technological University, Singapore
VinAI Research, Vietnam
anhtuan.luu@ntu.edu.sg

**Rongrong Ji** [*]
Xiamen University, China
rrji@xmu.edu.cn

**Hong Liu**
National Institute of Informatics, Japan
hliu@nii.ac.jp

## Abstract

Robustness against word substitutions has a well-defined and widely acceptable form, *i.e.*, using semantically similar words as substitutions, and thus it is considered as a fundamental stepping-stone towards broader robustness in natural language processing. Previous defense methods capture word substitutions in vector space by using either $l_2$-ball or hyper-rectangle, which results in perturbation sets that are not inclusive enough or unnecessarily large, and thus impedes mimicry of worst cases for robust training. In this paper, we introduce a novel *Adversarial Sparse Convex Combination* (ASCC) method. We model the word substitution attack space as a convex hull and leverages a regularization term to enforce perturbation towards an actual substitution, thus aligning our modeling better with the discrete textual space. Based on the ASCC method, we further propose ASCC-defense, which leverages ASCC to generate worst-case perturbations and incorporates adversarial training towards robustness. Experiments show that ASCC-defense outperforms the current state-of-the-arts in terms of robustness on two prevailing NLP tasks, *i.e.*, sentiment analysis and natural language inference, concerning several attacks across multiple model architectures. Besides, we also envision a new class of defense towards robustness in NLP, where our robustly trained word vectors can be plugged into a normally trained model and enforce its robustness without applying any other defense techniques. [1]

## 1 Introduction

Recent extensive studies have shown that deep neural networks (DNNs) are vulnerable to adversarial attacks (Szegedy et al., 2013; Goodfellow et al., 2015; Papernot et al., 2016a; Kurakin et al., 2017; Alzantot et al., 2018); *e.g.*, minor phrase modification can easily deceive Google's toxic comment detection systems (Hosseini et al., 2017). This raises grand security challenges to advanced natural language processing (NLP) systems, such as malware detection and spam filtering, where DNNs have been broadly deployed (Stringhini et al., 2010; Kolter & Maloof, 2006). As a consequence, the research on defending against natural language adversarial attacks has attracted increasing attention.

Existing adversarial attacks in NLP can be categorized into three folds: *(i)* character-level modifications (Belinkov & Bisk, 2018; Gao et al., 2018; Eger et al., 2019), *(ii)* deleting, adding, or swapping words (Liang et al., 2017; Jia & Liang, 2017; Iyyer et al., 2018), and *(iii)* word substitutions using semantically similar words (Alzantot et al., 2018; Ren et al., 2019; Zang et al., 2020). The first two attack types usually break the grammaticality and naturality of the original input sentences, and thus can be detected by spell or grammar checker (Pruthi et al., 2019). In contrast, the third attack type only substitutes words with semantically similar words, thus preserves the syntactic and semantics

---

[*]Corresponding author.
[1]Our code will be available at https://github.com/dongxinshuai/ASCC.

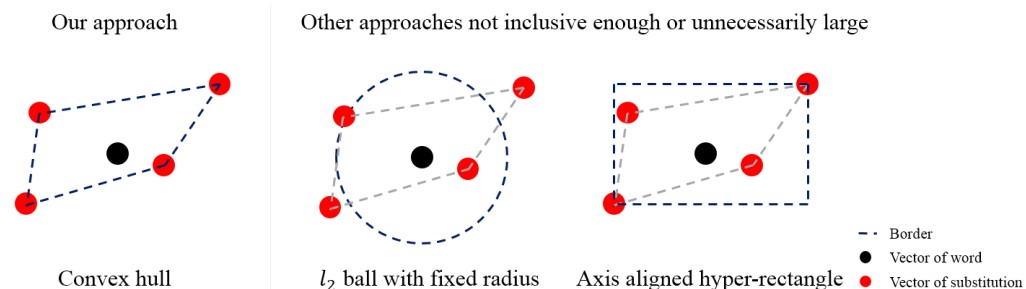

Figure 1: Visualization of how different methods capture the word substitutions in the vector space.

of the original input to the most considerable extent and are very hard to discern, even from a human's perspective. Therefore, building robustness against such word substitutions is a fundamental stepping stone towards robustness in NLP, which is the focus of this paper.

Adversarial attack by word substitution is a combinatorial optimization problem. Solving this problem in the discrete textual space is considered NP-hard as the searching space increases exponentially with the length of the input. As such, many methods have been proposed to model word substitutions in the continuous word vector space (Sato et al., 2018; Gong et al., 2018; Jia et al., 2019; Huang et al., 2019), so that they can leverage the gradients generated by a victim model either for attack or robust training. However, previous methods capture word substitutions in the vector space by using either $l_2$-ball or hyper-rectangle, which results in perturbation sets that are not inclusive enough or unnecessarily large, and thus impedes precise mimicry of the worst cases for robust training (see Fig. 1 for an illustration).

In this paper, we introduce a novel *Adversarial Sparse Convex Combination* (ASCC) method, whose key idea is to model the solution space as a convex hull of word vectors. Using a convex hull brings two advantages: (*i*) a continuous convex space is beneficial for gradient-based adversary generation, and (*ii*) the convex hull is, by definition, the smallest convex set that contains all substitutions, thus is inclusive enough to cover all possible substitutions while ruling out unnecessary cases. In particular, we leverage a regularization term to encourage adversary towards an actual substitution, which aligns our modeling better with the discrete textual space. We further propose ASCC-defense, which employs the ASCC to generate adversaries and incorporates adversarial training to gain robustness.

We evaluate ASCC-defense on two prevailing NLP tasks, *i.e.*, sentiment analysis on IMDB and natural language inference on SNLI, across four model architectures, concerning two common attack methods. Experimental results show that our method consistently yields models that are more robust than the state-of-the-arts with significant margins; *e.g.*, we achieve 79.0% accuracy under Genetic attacks on IMDB while the state-of-the-art performance is 75.0%. Besides, our robustly trained word vectors can be easily plugged into standard NLP models and enforce robustness without applying any other defense techniques, which envisions a new class of approach towards NLP robustness. For instance, using our pre-trained word vectors as initialization enhances a normal LSTM model to achieve 73.4% robust accuracy, while the state-of-the-art defense and the undefended model achieve 72.5% and 7.9%, respectively.

## 2 Preliminaries

### 2.1 Notations and Problem Setting

In this paper, we focus on text classification problem to introduce our method, while it can also be extended to other NLP tasks. We assume we are interested in training classifier $\mathcal{X} \rightarrow \mathcal{Y}$ that predicts label $y \in \mathcal{Y}$ given input $x \in \mathcal{X}$. The input $x$ is a textual sequence of $L$ words $\{x_i\}_{i=1}^{L}$. We consider the most common practice for NLP tasks where the first step is to map $x$ into a sequence of vectors in a low-dimensional embedding space, which is denoted as $v(x)$. The classifier is then formulated as $p(y|v(x))$, where $p$ can be parameterized by using a neural network, *e.g.*, CNN or LSTM model.

We examine the robustness of a model against adversarial word substitutions (Alzantot et al., 2018; Ren et al., 2019). Specifically, any word $x_i$ in $x$ can be substituted with any word $\hat{x}_i$ in $\mathbb{S}(x_i) = \{\mathbb{S}(x_i)_j\}_{j=1}^T$, where $\mathbb{S}(x_i)$ represents a predefined substitution set for $x_i$ (including itself) and $T$ denotes the number of elements in $\mathbb{S}(x_i)$. To ensure that $\hat{x}$ is likely to be grammatical and has the same label as $x$, $\mathbb{S}(x_i)$ is often comprised of semantically similar words of $x_i$, *e.g.*, its synonyms. Attack algorithms such as Genetic attack (Alzantot et al., 2018) and PWWS attack (Ren et al., 2019) aim to find the worst-case $\hat{x}$ to fool a victim model, whereas our defense methods aim to build robustness against such substitutions.

## 2.2 PERTURBATION SET AT VECTOR LEVEL

Gradients provide crucial information about a victim model for adversary generation (Szegedy et al., 2013; Goodfellow et al., 2015). However, in NLP, the textual input space is neither continuous nor convex, which impedes effective use of gradients. Therefore, previous methods capture perturbations in the vector space instead, by using the following simplexes (see Fig.1 for an illustration):

**$L_2$-ball with a fixed radius.** Miyato et al. (2017) first introduced adversarial training to NLP tasks. They use a $l_2$-ball with radius $\epsilon$ to constrain the perturbation, which is formulated as:

$$\hat{v}(x) = v(x) + r, \text{ s.t. } \|r\|_2 \leq \epsilon, \tag{1}$$

where $r$ denotes sequence-level perturbation in the word vector space and $\hat{v}$ denotes the adversarial sequence of word vectors. While such modeling initially considers $l_2$-ball at the sentence-level, it can also be extended to word-level to capture substitutions. Following that, Sato et al. (2018) and Barham & Feizi (2019) propose to additionally consider the directions towards each substitution. However, they still use the $l_2$-ball, which often fails to capture the geometry of substitutions precisely.

**Axis aligned bounds.** Jia et al. (2019) and Huang et al. (2019) use axis-aligned bound to capture perturbations at the vector level. They consider the smallest axis-aligned hyper-rectangular that contains all possible substitutions. Such perturbation set provides useful properties for bound propagation towards robustness. However, the volume of the unnecessary space it captures can grow with the depth of the model and grow exponentially with the dimension of the word vector space. Thus it fits shallow architectures but often fails to utilize the capacity of neural networks fully.

Besides, instead of fully defining the vector-level geometry of substitutions, Ebrahimi et al. (2018) propose to find substitutions by first-order approximation using directional gradients. It is effective in bridging the gap between continuous embedding space and discrete textual space. However, it is based on local approximation, which often fails to find global worst cases for robust training.

## 3 METHODOLOGY

In this section, we first introduce the intuition of using a convex hull to capture substitutions. Then, we propose how Adversarial Sparse Convex Combination (ASCC) generates adversaries. Finally, we introduce ASCC-defense that incorporates adversarial training towards robustness.

### 3.1 OPTIMALITY OF USING CONVEX HULL

From the perspective of adversarial defense, it is crucial to well capture the attack space of word substitutions. There are three aspects we need to consider: (i) *Inclusiveness*: the space should include all vectors of allowed substitutions to cover all possible cases. (ii) *Exclusiveness*: on the basis of satisfying inclusiveness, the space should be as small as possible since a loose set can generate unnecessarily intricate perturbations, which impede a model from learning useful information. (iii) *Optimization*: the space should be convex and continuous to facilitate effective gradient-based optimization, whether the objective function is convex or not (Bertsekas, 1997; Jain & Kar, 2017). Inspired by archetypal analysis (Cutler & Breiman, 1994), we propose to use a convex hull to build the attack space: the convex hull is a continuous space and, by definition, the minimal convex set containing all vectors of substitutions. We argue that using a convex hull can satisfy all the above aspects (as illustrated in Fig.1), and thus it is considered as theoretical optimum.

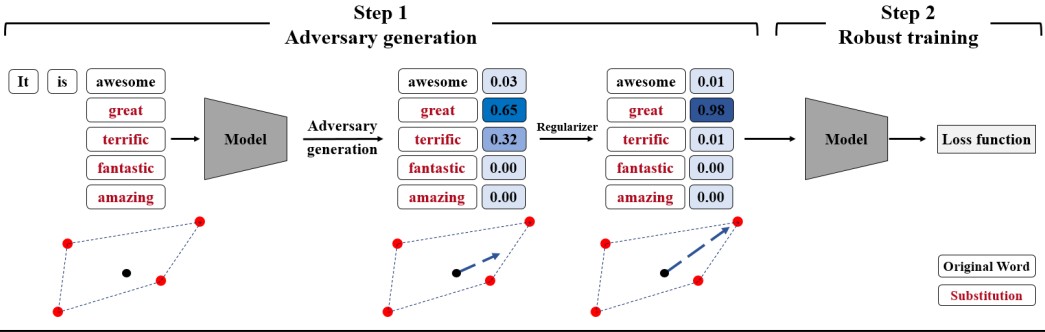

Figure 2: An illustration of the training process of the ASCC-defense. Step 1: Generate adversaries by ASCC with regularization. Step 2: Take adversaries as input to perform adversarial training.

### 3.2 ADVERSARIAL SPARSE CONVEX COMBINATION

**Efficient representation of and optimization over a convex hull.** A classical workaround in literature for optimization over a constraint set is the projected gradient descent (Cauchy, 1847; Frank & Wolfe, 1956; Bubeck, 2014; Madry et al., 2018). As for optimization over a convex hull, it necessitates characterizing the convex hull, *e.g.*, by vertexes, to perform projections. However, computing vertexes is computationally unfavorable because we need to recalculate the vertexes whenever word embeddings change, which frequently occurs during the training process.

In this paper, we propose a more efficient fashion for optimization over the concerning convex hull, based on the following proposition (the proof of which lies in the definition of convex hull):

**Proposition 1.** *Let* $\mathbb{S}(u) = \{\mathbb{S}(u)_1, ..., \mathbb{S}(u)_T\}$ *be the set of all substitutions of word* $u$, $\mathrm{conv}\mathbb{S}(u)$ *be the convex hull of word vectors of all elements in* $\mathbb{S}(u)$, *and* $v(\cdot)$ *be the word vector function. Then, we have* $\mathrm{conv}\mathbb{S}(u) = \{\sum_{i=1}^{T} w_i v(\mathbb{S}(u)_i) \mid \sum_{i=1}^{T} w_i = 1, w_i \geq 0\}$.

According to Proposition 1, we can formulate $\hat{v}(x_i)$, which denotes any vector in the convex hull around $v(x_i)$, as:

$$\hat{v}(x_i) = \sum_{j=1}^{T} w_{ij} v(\mathbb{S}(x_i)_j), \ \ \mathrm{s.t.} \sum_{j=1}^{T} w_{ij} = 1, w_{ij} \geq 0. \tag{2}$$

As such, we use Eq.2 to transform the original optimization on $\hat{v}(x_i)$ to the optimization on $w_i$, the coefficient of convex combination. Considering that $w_i$ still belongs to a set with constraint $\{\|w_i\|_1 = 1, w_{ij} \geq 0\}$, to achieve better flexibilities of optimization, we introduce a variable $\hat{w} \in \mathbb{R}$ to relax the constraint on $w$ by the following equation:

$$w_{ij} = \frac{\exp(\hat{w}_{ij})}{\sum_{j=1}^{T} \exp(\hat{w}_{ij})}, \hat{w}_{ij} \in \mathbb{R}. \tag{3}$$

After such relaxation in Eqs.2 and 3, we are able to optimize the objective function over the convex hull by optimizing $\hat{w} \in \mathbb{R}$. It provides a projection-free way to generate any adversaries inside the convex hull using gradients. .

**Gradient-based adversary generation.** Let $\mathcal{L}$ be a loss function concerning a classifier. We can generate the worst-case convex combinations $\hat{v}(x)$ by finding the worst-case $\hat{w}$:

$$\max_{\hat{w}} \mathcal{L}(v(x), \hat{v}(x), y) \tag{4}$$

where $\mathcal{L}$ is classification-related, *e.g.*, the cross-entropy loss over $\hat{v}(x)$:

$$\mathcal{L}(v(x), \hat{v}(x), y) = -\log p(y|\hat{v}(x)). \tag{5}$$

However, since we relax the discrete textual space to a convex hull in the vector space, any $w_i$ that $\|w_i\|_0 > 1$ is highly possible to give rise to $\hat{v}(x_i)$ that does not correspond to a real substitution.

---

**Algorithm 1** ASCC-defense

---

**Input**: dataset $\mathcal{D}$, parameters of Adam optimizer.
**Output**: parameters $\theta$ and $\phi$.

1: **repeat**
2:    **for** random mini-batch $\sim \mathcal{D}$ **do**
3:       **for** every $x$, $y$ in the mini-batch (in parallel) **do**
4:          Solve the inner maximization in Eq.11 to find the optimal $\hat{w}$ by Adam;
5:          Compute $\hat{v}(x)$ by Eq.10 using $\hat{w}$ and then compute the inner-maximum in Eq.11;
6:       **end for**
7:       Update $\theta$ and $\phi$ by Adam to minimize the calculated inner-maximum;
8:    **end for**
9: **until** the training converges.

---

To align better with the discrete nature of textual input, we propose to impose a regularizer on the coefficient of convex combination, $w_i$. To be specific, we take $w_i$ as a probability distribution and minimize the entropy function of $w_i$ to softly encourage the $l_0$ sparsity of $w_i$. We formulate this word-level entropy-based regularization term as:

$$\mathcal{H}(w_i) = \sum_{j=1}^{T} -w_{ij} \log(w_{ij}). \tag{6}$$

Combining loss function $\mathcal{L}$ and the entropy-based regularizer $\mathcal{H}$, we here formulate *Adversarial Sparse Convex Combination* (ASCC) for adversary generation as:

$$\max_{\hat{w}} \mathcal{L}(v(x), \hat{v}(x), y) - \alpha \sum_{i=1}^{L} \frac{1}{L} \mathcal{H}(w_i), \tag{7}$$

where $\alpha \geq 0$ is the weight controlling the regularization term (the effectiveness of which is validated in Sec.4.3).

### 3.3 ASCC-DEFENSE TOWARDS ROBUSTNESS

We here introduce ASCC-defense, which uses ASCC for adversaries and employs adversarial training towards robustness. We denote $\theta$ and $\phi$ as the parameters of $p(y|v(x))$ and $v(x)$, respectively.

**Adversarial training paradigm for NLP.** Adversarial training (Szegedy et al., 2013; Goodfellow et al., 2015; Madry et al., 2018) is currently one of the most effective ways to build robustness. Miyato et al. (2017) are the first to use adversarial training for text classification. They use $l_2$-ball with radius $\epsilon$ to restrict perturbations and the training objective can be defined as:

$$\min_{\theta, \phi} [\mathbb{E}_{(x,y) \sim \mathcal{D}} [\max_{r} \mathcal{L}(v(x), \hat{v}(x), y, \theta, \phi)]], \quad \text{s.t. } \hat{v}(x) = v(x) + r, \|r\|_2 \leq \epsilon, \tag{8}$$

where $r$ denotes the perturbations in the vector space and $\mathcal{L}$ denotes a classification-related loss. Therefore, maximizing $\mathcal{L}$ can generate adversarial perturbations $r$ to fool a victim model, whereas minimizing $\mathcal{L}$ can let the model learn to predict under perturbations.

**ASCC-Defense.** Instead of using $l_2$-ball in Eq.8, we leverage ASCC to capture perturbations inside the convex hull to perform adversarial training. This is to re-define $\hat{v}(x)$ in Eq.8 using ASCC, and the resulting training objective is formulated as:

$$\min_{\theta, \phi} [\mathbb{E}_{(x,y) \sim \mathcal{D}} [\max_{\hat{w}} \mathcal{L}(v(x), \hat{v}(x), y, \theta, \phi)]], \tag{9}$$

$$\hat{v}(x_i) = \sum_{j=1}^{T} w_{ij} v(\mathbb{S}(x_i)_j), \ w_{ij} = \frac{\exp(\hat{w}_{ij})}{\sum_{j=1}^{T} \exp(\hat{w}_{ij})}. \tag{10}$$

To specify $\mathcal{L}$ in Eq.9 for ASCC-defense, we consider the KL-divergence between the prediction by vanilla input and the prediction under perturbations (Miyato et al., 2018; Zhang et al., 2019a). In the meantime, we also encourage the sparsity of $w_i$ by the proposed regularizer for adversary generation. Taking these together, we formulate the training objective of ASCC-defense as follows:

$$\min_{\theta, \phi} [\mathbb{E}_{(x,y) \sim \mathcal{D}} [\max_{\hat{w}} -\log p(y|v(x)) - \alpha \sum_{i=1}^{L} \frac{1}{L} \mathcal{H}(w_i) + \beta \text{KL}(p(\cdot|v(x))||p(\cdot|\hat{v}(x))) ]], \tag{11}$$

Table 1: Accuracy(%) of different defense methods under attacks on IMDB (a) and SNLI (b). "First-order aprx" denotes Ebrahimi et al. (2018). "Adv $l_2$-ball" denotes Miyato et al. (2017). "Axis-aligned" denotes Jia et al. (2019). "ASCC-defense" denotes the proposed method.

| Method | Model | Genetic | PWWS | Method | Model | Genetic | PWWS |
|---|---|---|---|---|---|---|---|
| Standard | LSTM | 1.0 | 0.2 | Standard | BOW | 28.8 | 15.4 |
| | CNN | 7.0 | 11.3 | | DCOM | 30.2 | 9.0 |
| First-order aprx | LSTM | 72.5 | 66.7 | First-order aprx | BOW | 65.6 | 57.2 |
| | CNN | 51.2 | 74.1 | | DCOM | 66.7 | 58.6 |
| Adv $l_2$-ball | LSTM | 20.1 | 11.7 | Adv $l_2$-ball | BOW | 35.0 | 16.7 |
| | CNN | 36.7 | 46.2 | | DCOM | 33.1 | 15.4 |
| Axis-aligned | LSTM | 64.7 | 59.6 | Axis-aligned | BOW | 75.0 | 72.1 |
| | CNN | 75.0 | 69.5 | | DCOM | 73.7 | 67.9 |
| ASCC-defense | LSTM | **79.0** | **77.1** | ASCC-defense | BOW | **76.3** | **75.1** |
| | CNN | **78.2** | **76.2** | | DCOM | **74.5** | **72.8** |


(a) Accuracy (%) under attacks on IMDB.
(b) Accuracy (%) under attacks on SNLI.


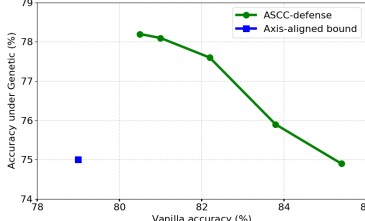
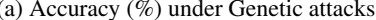

(a) Accuracy (%) under Genetic attacks.

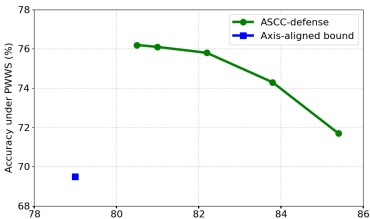

(b) Accuracy (%) under PWWS attacks.

Figure 3: Tradeoff between robustness and accuracy on IMDB under Genetic and PWWS attacks.

where $\alpha, \beta \geq 0$ control the weight of regularization and KL term, respectively. Noted that term $\mathcal{H}(\cdot)$ has no gradient with respect to $\theta$ and $\phi$, so it only works during inner-max adversary generation.

**Robust word vectors.** We here explain why ASCC-defense can yield more robust word vectors. Previous defenses such as Miyato et al. (2017) fail to train word vectors in a robust way, as they update $\phi$ by only using the clean data flow. Specifically, Miyato et al. (2017) obtains $\hat{v}(x_i)$ through Eq.1, where perturbation $r$ has no gradient with respect to $\phi$, and thus $\nabla_\phi \hat{v}(x_i) = \nabla_\phi v(x_i)$. On the contrary, $\hat{v}(x_i)$ modeled by ASCC-defense has gradient *w.r.t.* $\phi$ concerning all substitutions, as:

$$\nabla_\phi \hat{v}(x_i) = \sum_{j=1}^{T} w_{ij} \, \nabla_\phi v(S(x_i)_j). \tag{12}$$

Therefore, ASCC-defense updates the word vector considering all potential adversarial substitutions simultaneously, which gives rise to more robust word vectors (we validate our claim in Sec.4.4).

**Optimization.** We employ Adam (Kingma & Ba, 2014) to solve both inner-max and outer-min problems in Eq.11. Our training process is illustrated in Fig. 2 and presented in Algorithm 1.

## 4 EXPERIMENTS

### 4.1 EXPERIMENTAL SETTING

**Tasks and datasets.** We focus on two prevailing NLP tasks to evaluate the robustness and compare our method to the state-of-the-arts: (i) Sentiment analysis on the IMDB dataset (Maas et al., 2011). (ii) Natural language inference on the SNLI dataset (Bowman et al., 2015).

**Model architectures.** We examine robustness on the following four architectures to show the scalability of our method: (i) BOW, the bag-of-words model which sums up the word vectors and predicts

Table 2: Ablation study on the sparsity regularization term.

| Regul weight | Vanilla | Genetic | PWWS | | Regul weight | Vanilla | Genetic | PWWS |
|---|---|---|---|---|---|---|---|---|
| $\alpha$=0 | 80.6 | 75.1 | 61.6 | | $\alpha$=0 | 76.7 | 73.4 | 73.7 |
| $\alpha$=5 | 81.9 | 76.8 | 71.7 | | $\alpha$=5 | 77.4 | 75.1 | 74.0 |
| **$\alpha$=10** | 82.2 | 78.2 | 75.7 | | **$\alpha$=10** | 77.8 | 76.3 | 75.1 |
| $\alpha$=15 | 81.2 | 76.1 | 78.3 | | $\alpha$=15 | 76.7 | 73.7 | 73.3 |

(a) Acc (%) of CNN-based ASCC-defense on IMDB.  (b) Acc (%) of BOW-based ASCC-defense on SNLI.

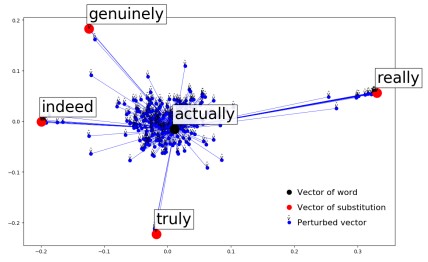

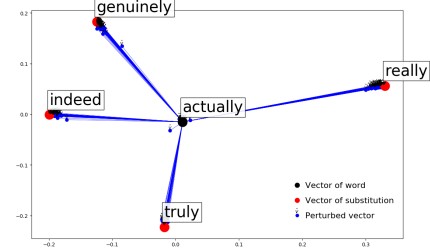

(a) $\hat{v}$ generated without the regularization term.  (b) $\hat{v}$ generated with the regularization when $\alpha = 10$.

Figure 4: An illustration to show the effectiveness of the proposed sparsity regularization. We randomly choose 300 adversaries of word "actually" from the test set of IMDB. Vectors are projected into a 2-dimensional space by SVD. Best view in color with zooming in.

by a multilayer perceptron (MLP), (ii) CNN model, (iii) Bi-LSTM model, (iv) DCOM, decomposable attention model (Parikh et al., 2016), which generates context-aware vectors and predicts by a MLP. We align our implementation details with Jia et al. (2019) for fair comparisons.

**Comparative methods.** (i) Standard training. It uses the cross-entropy loss as the main loss function. (ii) First-order approximation (Ebrahimi et al., 2018). While it is initially proposed to model char flip, it can also be applied to word substitutions. We implement its word-level version for comparison. (iii) Adv $l_2$-ball (Miyato et al., 2017). It first normalizes the word vectors and then generates perturbations inside a $l_2$-ball for adversarial training. We implement it with word level $l_2$-ball and radius $\epsilon$ varying from 0.1 to 1. We only plot the best performance among different $\epsilon$. (iv) Axis aligned bound (Jia et al., 2019). It models perturbations by an axis-aligned box and uses bound propagation for robust training. (v) ASCC-defense. We set the hyper-parameters $\alpha = 10$ and $\beta = 4$. For fair comparisons, KL term is employed for all compared adversarial training based methods. More implementation details as well as runtime analysis can be found in the Appendix A.

**Attack algorithms.** We employ the following two powerful attack methods to examine the robustness: (i) Genetic attack (Alzantot et al., 2018) maintains a population to generate attacks in an evolving way. Aligned with Jia et al. (2019), we set the population size as 60 to run for 40 iterations. (ii) PWWS attack (Ren et al., 2019) calculates the saliency of each word and then substitutes greedily. Aligned with Alzantot et al. (2018) and Jia et al. (2019), we do not attack premise on SNLI.

**Substitution set.** For fair comparisons with state-of-the-art defense Jia et al. (2019), we follow their setting to use substitution set from Alzantot et al. (2018). We apply the same language model constraint on Genetic as in Jia et al. (2019), while do not apply it on PWWS attacks. As we aim at using prevailing setting to compare with the state-of-the-arts, we do not focus on how to construct the substitution set in this work and leave it for future exploration.

## 4.2 MAIN RESULT

Aligned with Jia et al. (2019), we evaluate robustness on 1000 randomly selected examples from the test set of IMDB, and all 9824 test examples from SNLI. As shown in Tab.1, our method achieves leading robustness across all architectures with significant margins. For example, on IMDB, we surpass LSTM-based runner-up method by 6.5% under Genetic and 10.4% under PWWS attacks.

Table 3: Accuracy (%) of models initialized with different word vectors *without* any other defense technique. GloVe denotes the word vectors from Pennington et al. (2014). "First order V" denotes word vectors trained by Ebrahimi et al. (2018). "ASCC-V" denotes word vectors trained by ASCC-defense. We freeze the pre-trained word vectors during normal training.

| Word vector | Model | Vanilla | Genetic | Word vector | Model | Vanilla | Genetic |
|---|---|---|---|---|---|---|---|
| GloVe | LSTM | **88.5** | 7.9 | GloVe | BOW | **80.1** | 35.8 |
| First order V | LSTM | 85.3 | 65.6 | First order V | BOW | 79.3 | 62.1 |
| ASCC-V | LSTM | 84.1 | **73.4** | ASCC-V | BOW | 77.9 | **69.6** |
| GloVe | CNN | **86.4** | 8.6 | GloVe | DCOMP | **82.6** | 41.8 |
| First order V | CNN | 83.1 | 44.7 | First order V | DCOMP | 78.7 | 62.8 |
| ASCC-V | CNN | 84.2 | **72.0** | ASCC-V | DCOMP | 77.8 | **72.1** |

(a) Accuracy (%) under attacks on IMDB.      (b) Accuracy (%) under attacks on SNLI.

Plus, the robust performance of ASCC-defense is consistent against different attacks: *e.g.*, on IMDB, LSTM-based ASCC-defense achieves 79.0% under Genetic attacks and 77.1% under PWWS attacks, which shows ASCC-defense does not rely on over-fitting to a specific attack algorithm. In addition to robust accuracy, we also plot the tradeoff between robustness and accuracy in Fig.3, which shows our method can trade off some accuracy for more robustness compared to the state-of-the-art. More detailed vanilla accuracy and our performance on BERT (Devlin et al., 2019) are shown in Appendix B.

### 4.3 ON THE REGULARIZATION AND OTHER DISCUSSIONS

Fig.4 qualitatively shows how the proposed regularizer encourages sparsity. After applying the regularization, the resulting $\hat{v}$ is close to a substitution, corresponding better with the discrete nature of textual input. Tab.2 quantitatively shows the influence of the regularization term on robustness. Specifically, when $\alpha = 10$ our method performs the best. As $\alpha$ keeps increasing, ASSC focus too much on the sparsity and thus fail to find strong enough perturbations for robust training.

We now discuss some other intuitive defense methods. The thought of enumerating all combinations during training is natural and yet impractical on benchmark datasets; *e.g.*, on IMDB the average number of combinations per input is $6^{108}$. Augmenting training with random combinations is also ineffective, since it fails to find hard cases in the exponentially large attack space; *e.g.*, under Genetic ASCC-defense surpasses random augmentation by 46.0% on IMDB and by 7.8% on SNLI (more significant margin owes to larger attack space on IMDB). Besides, though simply grouping all substitutions can achieve ensured robustness, it sacrifices discriminative powerness: two words that are not semantically similar will be mapped together just because they are indirectly related by one or more mediators. For instance, grouping defense achieves 71.3% robust accuracy and 71.3% vanilla accuracy on IMDB while ASCC-defense achieves 79.0% and 82.5% respectively.

### 4.4 ROBUST WORD VECTORS

As mentioned in Sec.3.3, ASCC-defense updates the vector of a word by considering all its substitutions, and thus the obtained word vectors are robust in nature. To validate, we use the standard training process to train models but with different pre-trained word vectors as initialization. We compare word vectors pre-trained by ASCC-defense with Glove and word vectors pre-trained by Ebrahimi et al. (2018) (the best performing setting for Miyato et al. (2017) and Jia et al. (2019) is to freeze the word vectors as Glove, which laterally validates our claim about robust word vectors in Sec.3.3). As shown in Tab.3, the models initialized by our robustly trained word vectors (and fixed during normal training) are robust to attacks without applying any other defense techniques. For example, armed with our robust word vectors, a normally trained LSTM-based model can achieve 73.4% under Genetic attacks, whereas using GloVe achieves 7.9%.

In addition, this result also implies a new perspective towards robustness in NLP: the vulnerabilities of NLP models relate to word vectors significantly, and transferring pre-trained robust word vectors can be a more scalable way towards NLP robustness. For more result, please refer to Appendix B.

## 5 RELATED WORK

Though achieved success in many fields, DNNs appear to be susceptible to adversarial examples (Szegedy et al., 2013). Initially introduced to attack CV models, attack algorithms vary from $L_p$ bounded Goodfellow et al. (2015); Carlini & Wagner (2017); Madry et al. (2018), universal perturbations (Moosavi-Dezfooli et al., 2017; Liu et al., 2019), to wasserstein distance-based attack (Wong et al., 2019), while defense techniques for CV models include adversarial training(Goodfellow et al., 2015; Kurakin et al., 2017; Madry et al., 2018), preprocessing (Chen et al., 2018; Yang et al., 2019), and generative classifiers (Li et al., 2019; Schott et al., 2019; Dong et al., 2020).

Recently, various classes of NLP adversarial attacks have been proposed. Typical methods consider char-level manipulations (Hosseini et al., 2017; Ebrahimi et al., 2018; Belinkov & Bisk, 2018; Gao et al., 2018; Eger et al., 2019; Pruthi et al., 2019). Another line of thought focus on deleting, adding, or swapping words (Iyyer et al., 2018; Ribeiro et al., 2018; Jia & Liang, 2017; Zhao et al., 2018).

In contrast to char-level and sequence-level manipulations, word substitutions consider prior knowledge to preserve semantics and syntactics, such as synonyms from WordNet (Miller, 1998), Sememes (Bloomfield; Dong et al., 2006), and neighborhood relationships (Alzantot et al., 2018), Some focus on heuristic searching in the textual space (Alzantot et al., 2018; Liang et al., 2017; Ren et al., 2019; Jin et al., 2020; Zhang et al., 2019b; Zang et al., 2020), while (Papernot et al., 2016b; Gong et al., 2018) propose to leverage gradients for adversary generation in the vector space.

As for defense against adversarial word substitutions, Ebrahimi et al. (2018) find substitutions by first-order approximation. Miyato et al. (2017), Barham & Feizi (2019) and Sato et al. (2018) use $l_2$-ball to model perturbations, while Jia et al. (2019) and Huang et al. (2019) use axis-aligned bound. Zhou et al. (2020) sample from Dirichlet distribution to initialize a convex combination of substitutions, but the sparsity might be lost during adversary generation. Our work differs as we model the convex hull with sparsity by entropy function, and the sparsity is enforced during the whole process, which makes our captured geometry of substitutions more precise.

## 6 CONCLUSION

In this paper, we proposed a novel method to use the convex hull to capture and defense against adversarial word substitutions. Our method yields models that consistently surpass the state-of-the-arts across datasets and architectures. The experimental results further demonstrated that the word vectors themselves can be vulnerable and our method gives rise to robust word vectors that can enforce robustness without applying any other defense techniques. As such, we hope this work can be a stepping stone towards even broader robustness in NLP.

## ACKNOWLEDGEMENTS

This work is supported by the National Science Fund for Distinguished Young Scholars (No. 62025603), and the National Natural Science Foundation of China (No. U1705262, No. 62072386, No. 62072387, No. 62072389, No. 62002305, No. 61772443, No. 61802324, and No. 61702136).

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

## A  APPENDIX

### A.1  IMPLEMENTATION DETAILS

**Text processing.** We employ the tokenizer from NLTK and ignore all punctuation marks when tokenizing the textual input. We set the maximum length of input as 300 for IMDB and 80 for SNLI. For unknown words, we set them as "NULL" token.

**Hyper-parameters and optimization.** We set $\alpha$ as 10 and $\beta$ as 4 for the training procedure defined in Eq.12. To generate adversaries for robust training, we employ Adam optimizer with a learning rate of 10 and a weight decay of 0.00002 to run for 10 iterations To update $\phi$ and $\theta$, we also employ Adam optimizer, the parameters of which differ between architectures and will be discussed as follows.

**Architecture parameters** (i) CNN for IMDB: We use a 1-d convolutional layer with kernal size of 3 to extract features and then make predictions. We set the batch-size as 64 and use Adam optimizer with a learning rate of 0.005 and a weight decay of 0.0002. (ii) Bi-LSTM for IMDB: We use a bi-directional LSTM layer to process the input sequence, and then use the last hidden state to make predictions. We set the batch-size as 64 and use Adam optimizer with a learning rate of 0.005 and a weight decay of 0.0002. (iii) BOW for SNLI: We first sum up the word vectors at the dimension of sequence and concat the encoding of the premise and the hypothesis. Then we employ a MLP of 3 layers to predict the label. We set the batch-size as 512 and use Adam optimizer with a learning rate of 0.0005 and a weight decay of 0.0002. (iv) DECOMPATTN for SNLI: We first generates context-aware vectors and then employ a MLP of 2 layers to make predictions given the context-aware vectors. We set the batch-size as 256 and use Adam with a learning rate of 0.0005 and a weight decay of 0.

### A.2  RUNTIME ANALYSIS

All models are trained using the GeForce GTX1080 GPU. (i) As for IMDB, it takes about 1.5 GPU hours to train a CNN-based model and 2 GPU hours for a LSTM-based model. (ii) As for SNLI, it takes about 12 GPU hours to train a BOW-based model and 15 GPU hours for DECOMPATTN-based model.

Table 4: Vanilla accuracy(%) of different defense methods on IMDB (a) and SNLI (b).

| Method | Model | Vanilla accuracy |
|---|---|---|
| Standard | LSTM | 88.5 |
| | CNN | 87.2 |
| First-order aprx | LSTM | 83.2 |
| | CNN | 80.3 |
| Adv $l_2$-ball | LSTM | 84.6 |
| | CNN | 84.5 |
| Axis-aligned | LSTM | 76.8 |
| | CNN | 81.0 |
| ASCC-defense | LSTM | 82.5 |
| | CNN | 81.7 |

(a) Vanilla accuracy (%) on IMDB.

| Method | Model | Vanilla accuracy |
|---|---|---|
| Standard | BOW | 80.1 |
| | DECOMP | 82.6 |
| First-order aprx | BOW | 78.2 |
| | DECOMP | 77.6 |
| Adv $l_2$-ball | BOW | 74.8 |
| | DECOMP | 73.5 |
| Axis-aligned | BOW | 79.4 |
| | DECOMP | 77.1 |
| ASCC-defense | BOW | 77.2 |
| | DECOMP | 76.3 |

(b) Vanilla accuracy (%) on SNLI.

Table 5: Vanilla and robust accuracy (%) of the proposed method on BERT (bert-base-uncased).

| Method | Dataset | Model | Vanilla accuracy | Under Genetic attack |
|---|---|---|---|---|
| Standard | IMDB | BERT | 92.2 | 16.4 |
| ASCC-defense | IMDB | BERT | 77.5 | 70.2 |

## B  ADDITIONAL EXPERIMENTAL RESULT

### B.1  VANILLA ACCURACY AND ROBUST ACCURACY UNDER GENETIC ATTACKS WITHOUT CONSTRAINT

In Tab.1, we have plotted the robust accuracy under attacks to compare with state-of-the-arts. Here we plot the vanilla accuracy of all compared methods in Tab.4 as an addition. Aligned with Tab.3, the parameters of each compared method here are chosen to have the best robust accuracy instead of vanilla accuracy.

In our main result in Tab.1, for fair comparisons we align our setting with SOTA defense Jia et al., where genetic attacks are constrained by a language model. Here we plot our performance under genetic attacks without any language model constraint as an addition: ASCC-defense achieves 76.7% robust accuracy under Genetic without constraint on IMDB based on LSTM, and 72.8% on SNLI based on BOW.

### B.2  PERFORMANCE ON BERT

ASCC-defense models perturbations at word-vecotr level, and thus it can be applied to architectures like Transformers, as long as it uses word embeding as its first layer. To validate, we conduct experiments on BERT (bert-base-uncased) using standard training and ASCC-defense respectively. As shown in Tab.5, ASCC-defense enhances the robustness of BERT model significantly. Specifically, BERT finetuned by standard method on IMDB achieves 16.4% robust accuracy under Genetic attacks, while using the proposed ASCC-defense achieves 70.2%.

Table 6: Accuracy (%) of normally trained models initialized with and freezed by the proposed robust word vectors. For example, pre-trained on LSTM means the word vectors are pre-trained by LSTM-based ASCC-defense and applied to CNN means the pre-trained word vectors are used to initialize a CNN model to perform normal training.

| Pre-trained | Applied to | Vanilla | Genetic |
|---|---|---|---|
| LSTM | LSTM | 84.1 | 73.4 |
| LSTM | CNN | 78.5 | 71.9 |

(a) Accuracy (%) under attacks on IMDB.

| Pre-trained | Applied to | Vanilla | Genetic |
|---|---|---|---|
| DECOMP | DECOMP | 77.8 | 72.1 |
| DECOMP | BOW | 77.2 | 70.5 |

(b) Accuracy (%) under attacks on SNLI.

### B.3 REDUCED PERTURBATION REGION

In this section, we show the reduced perturbation region by using convex hull compared to $l_2$-ball and hyper-rectangle. To make the result more intuitive, we first project word vectors into a 2D space by SVD, and than calculate the average area of each modeling (to rule out irrelevant factors, we use word vectors from GloVe and consider words whose substitution sets are of the same size). We choose the smallest $l_2$-ball and hyper-rectangle that contain all substitutions to compare with convex hull. The result shows that using convex hull reduce the perturbation region significantly. Specificaly, the average ratio of the area modeled by convex hull to the area modeled by hyper-rectangle is 29.2%, and the average ratio of the area modeled by convex hull to the area modeled by $l_2$-ball area is 8.4%.

### B.4 CROSS-ARCHITECTURES PERFORMANCE OF ROBUST WORD VECTORS

As discussed in Sec.4.4, our robustly trained word vectors can enforce the robustness of a normally trained model without applying any other defense techniques. In this section, we aim to examine whether such a gain of robustness over-fits to a specific architecture. To this end, we first employ ASCC-defense to obtain robust word vectors and then fix the word vectors as the initialization of another model based on a different architecture to perform normal training. We examine the accuracy under attacks. As shown in Tab.6, our robustly trained word vectors consistently enhance the robustness of a normally trained model based on different architectures. For example, though trained by a LSTM-based model, the robust word vectors can still enforce a CNN-based model to achieve robust accuracy of $71.9\%$ under Genetic attacks (whereas initializing by GloVe achieves $8.6\%$), demonstrating the across-architecture transferability of the robustness of our pre-trained word vectors .

