# OpenReview forum: "Towards Robustness Against Natural Language Word Substitutions"
_ICLR.cc/2021/Conference — ICLR 2021 Spotlight_

### Official Review · AnonReviewer2 · 2020-10-27
**simple idea and strong results**

**Rating:** 7
**Confidence:** 3

**Review:**

*Summary of the paper*: This paper studies the problem of robustness against word substitutions. The authors propose a novel Adversarial Sparse Convex Combination (ASCC) method in which they model the word substitution attack space as a convex hull and leverages a regularization term to enforce perturbation towards an actual substitution. Based on the ASCC, they also propose ASCC-defense, which leverages ASCC to generate worst-case perturbations and incorporates adversarial training towards robustness. Experimental results show that their method outperforms the existing SOTA on two tasks -- sentiment analysis and natural language inference.

*Strength of the paper*:
1. The idea proposed in the paper is quite straightforward yet effective. Using a convex hull could satisfy the three aspects as stated by the authors -- Inclusiveness, Exclusiveness, and Optimization. The experimental results do show the advantage of using the proposed convex hull.

2. Besides the robustness of the model, it also achieves robustness over word vectors.

3. The experiments are well designed including both qualitative analysis, quantitative results, and reasonable ablation to show the effectiveness of their method. In general, the paper is well-structured and easy to follow.

*Question for the authors*:

1. I am curious about how much headroom there still exists for this task? Let's say if we have a perfect way to defend against such kind of attacks, how good are the current approaches?

2. What is the computational cost of the proposed method compared to others? In equation (3), the authors compute $w_{ij}$ through a softmax parameterized by $\hat{w_{ij}}$. If the substitution set of a word is infinite or too large, how are you going to deal with such kind of situations?

3. Seen from Table 1, the proposed method is much better than other models at PWWS attacks than the genetic attacks. Can you give an intuitive explanation of why?

*Reason for score*: Overall, I vote for accepting this paper. I like the idea of using a convex hull and the way they regularize the model to achieve sparsity. It would be helpful if the authors could address the questions raised above.

---

> ### Author Response · Authors · 2020-11-15
> **Author response**
>
> We thank the reviewer for the insightful review and valuable feedback.
>
> Regarding how much headroom there still exists: We think that this is an open problem in NLP robustness. The trade-off between standard accuracy and robust accuracy has been widely concerned in computer vision communities [1, 2]. We believe that the phenomenon of tradeoff also occurs in NLP tasks, though there is a lack of corresponding research. In this sense, many researchers hoped that the best robust accuracy can be as close to the standard accuracy as possible. For example, on IMDB, the vanilla accuracy using LSTM is 88.5% while the robust accuracy (under Genetic attack) is 79.0%, with a gap of nearly 10%. As such, there is still a ways to go, not to mention that attack algorithms also evolve with time.
>
> Regarding the computational cost, please refer to our runtime analysis in Appendix A.2. For example, on IMDB it takes about 1.5 hours for the proposed method to train on CNN and 2 hours on LSTM, while it takes 0.6 hours for Jia et al.’s method [3] to train on CNN and 29 hours on LSTM. Even when we expand the substitution set 10 times bigger to |S(u)|=200 (which is fairly large enough), the computational overhead is still acceptable; e.g., the training time on IMDB will be 15 hours on CNN and 20 hours on LSTM in this case.
>   If we consider other settings such as substitutions using words with the same POS to attack a grammar checker, the substitution set may go even larger. To solve this problem, we can first identify the most K representative words of a substitution set and then use these words to capture the convex hull approximately.
>
> Regarding why the proposed method is much better than other models under PWWS attacks than the genetic attacks, to align with SOTA defense [3], we apply the same language model constraint on genetic attack during evaluation. To consider a harder situation, we do not apply this constraint on PWWS attack (Section 4.1). Therefore, the PWWS attack might be more powerful during our evaluation (because we do not apply constraint on it) and it may partially explain why the proposed method is much better than other models under PWWS attack. We understand that you may favor an intuitive explanation considering the mechanism and nature of the two attacks. We therefore conducted additional experiments where we did not apply language model constraint on both attacks and found that the phenomenon is not as obvious. We will add this result to the revision as a supplement.
>
> Reference:
>
> [1] Tsipras, Dimitris, Shibani Santurkar, Logan Engstrom, Alexander Turner, and Aleksander Madry. Robustness may be at odds with accuracy. In ICLR, 2019.
>
> [2] Hongyang Zhang, Yaodong Yu, Jiantao Jiao, Eric P. Xing, Laurent El Ghaoui, and Michael I. Jordan. Theoretically principled trade-off between robustness and accuracy. In ICML, 2019a.
>
> [3] Robin Jia, Aditi Raghunathan, Kerem Goksel, and Percy Liang. Certified robustness to adversarial word substitutions. In EMNLP, 2019.

---

### Official Review · AnonReviewer4 · 2020-10-28
**Interesting paper. I only have some questions.**

**Rating:** 7
**Confidence:** 4

**Review:**

Summary:
In this paper, the authors aim to build a robust model against word substitution attacks. Unlike previous work, they consider a convex hull as the perturbation region instead of a norm-ball or a hyper-rectangle. From their derivation, perturbed words can be viewed as the linear combinations of substitutions and perturbations can be viewed as the corresponding normalized weights. Therefore, they can adversarially train the perturbations and model to obtain a robust model and robust word embeddings. The authors also design a regularizer to encourage the sparsity on perturbation weights. The experimental results show that the proposed model is indeed more robust than other baselines. In addition, they show that the learned word embedding can be a good initialization for training robust models.

I very like the idea to convert the perturbations into the linear combination weights. I only have some minor questions:
- Can the proposed method be applied to more complex models, such as Transformer?
-Can the proposed technique be extended to optimize the certification bound (the lower bound of robust accuracy)?
- Maybe it is interesting to report the percentage of reduced perturbation region by using the convex hull rather than norm-ball or hyper-rectangle.

---

> ### Author Response · Authors · 2020-11-15
> **Author response**
>
> We thank the reviewer for the insightful review and valuable feedback.
>
> Regarding the applicability of the proposed method to more complex models such as Transformer, our proposed method can be applied to any architecture as long as the architecture uses word embedding as its first layer, which is fairly common in current NLP models. We are running experiments on BERT and will update the results as a supplement.
>
> Regarding the question of whether the proposed method can be extended to optimize the certification bound, the answer is yes. As for bound propagation, the proposed method can be used for pretraining. As for certified robustness by smoothified model, the proposed method can be leveraged to improve the smoothing technique and we plan to leave it for future work.
>
> Regarding the percentage of reduced perturbation region, we thank the reviewer for the suggestion. We will report it in the revision.

---

> ### Author Response · Authors · 2020-11-19
> **Experiments on BERT**
>
> I would like to update our experimental result on BERT. On IMDB normally finetuned BERT (bert-base-uncased) achieves 16.4% robust accuracy under genetic attacks, while using our defense method achieves 70.2% robust accuracy. As we are still tuning hyper-parameters, our final result on BERT might be even higher.

---

### Official Review · AnonReviewer3 · 2020-11-02
**Interesting idea but a few questions**

**Rating:** 7
**Confidence:** 3

**Review:**

The authors answered my questions so I am increasing my score to 7.

-----

The paper presents a new defense for being robust to adversarial examples in NLP. This is a very exciting topic and I am glad to see more work in this space.

The paper presents a technique to make the model robust to word substitutions coming from a set S(u). The authors propose to use a convex hull which the authors claim is a better bound than using l2-ball or axis-aligned rectangles.

The authors derive a optimization objective for their problem and show experimental results that shows their model achieves higher performance under two types of attacks (Genetic and PWWS) on the IMDB and SNLI datasets for various standard neural architectures.


I have a couple of questions:

(1) Is this setting different than the one explored in Jia et al. 2019 (Certified Robustness to Adversarial Word Substitutions)?

It seems Jia et al. 2019 explores the case where multiple positions in the sentence can be perturbed whereas here only one word can be perturbed? I think clarifying this and discussing its implications would be useful for the reader.

(2) In training does the model have access to the set of all possible substitutions S(u) or not?

---

> ### Author Response · Authors · 2020-11-15
> **Author response**
>
> We thank the reviewer for the insightful review and valuable feedback.
>
> Regarding perturbation of multiple positions (question 1), we use the same setting as in Jia et al. 2019 (Certified Robustness to Adversarial Word Substitutions), where adversarial substitutions are allowed at multiple positions concurrently. At the premise of preserving the syntactic and semantics (e.g., using synonyms), substituting at multiple positions rather than single positions can generate more powerful attacks, and thus, defense under such a scenario is more practically meaningful. We will update the paper to clarify this and discuss its implication as the reviewer’s suggestion.
>
> Regarding the access to all possible substitution set (question 2), this setting is also aligned with SOTA defense Jia et al. 2019 (Certified Robustness to Adversarial Word Substitutions), i.e. the model have the access to the set of all possible substitutions during the training. This is for fair comparison between defense techniques as it rules out the effect of using different substitution set. Defense is not trival, though the model has access to the set of substitutions, as the attack space can be exponentially large; e.g., on IMDB, there are 6^108 possible combinations per input on average, which makes simple enumeration impractical and random augmentation ineffective (Section 4.3).

---

### Public Comment · ~Jiehang_Zeng1 · 2020-11-15
**A few questions**

This paper proposes an empirical approach to defending attacks based on word substitutions, and I have some questions:

- Zhou et al. [1] introduced an idea similar to your work:
(a) adversarial optimization in a convex hull, with the same method in this work.
(b) encouraging the sparsity of weights: they use Dirichlet distribution and you use entropy. In my opinion, I think your method is more elegant.
However, it seems that Zhou et al. and this work come to different conclusions: they think a sparse combination (Table 4 in [1]) contributes to higher clean accuracy and lower robust accuracy, while in this paper a larger $\alpha$ contributes to better robust accuracy.

- I am not sure whether the defender has full access to the synonym set used by the attacker. If so, this would limit the applicability of the method. Different attackers may use different synonym sets, e.g., PWWS extracts the synonym set from wordnet, TextFooler uses top-50 closest words in the embedding space, etc. If the overlap between the synonym set used by attackers and defenders is small, the result may be influenced.

- Have you tried to evaluate the result on more popular pretrained models such as BERT, RoBERTa, et al?  Zhou et al. [1] only conducts experiments with BERT on SNLI, while I think more SOTA models deserve exploration. One key issue may be how to apply embedding combination on models using word piece tokenization.

- Does logit pairing help improve robust accuracy? If so, can you show some ablation study of $\beta$?


References:
[1] Zhou et al. "Defense against adversarial attacks in nlp via dirichlet neighborhood ensemble." arXiv preprint arXiv:2006.11627 (2020).

---

> ### Author Response · Authors · 2020-11-18
> **Author response**
>
> Thank you for your interest in our work.
>
> We thank the reviewer for mentioning an interesting work from Zhou et al. ("Defense against Adversarial Attacks in Nlp via Dirichlet Neighborhood Ensemble." arXiv preprint arXiv:2006.11627 (2020)), and also for that you think our method is more elegant. In a nutshell, we capture the convex hull with sparsity by employing entropy function, which is very different from Zhou et al. Specifically, Zhou et al. sample from a Dirichlet distribution to initialize a sparse convex combination. However, during the process of adversary generation, the Dirichlet distribution is not considered anymore and the sparsity gained from the initialization phase may be lost. In contrast, our method considers the sparsity of convex combination during the whole process. We incorporate entropy function as a regularization term to the loss, whose gradients is used to guide the adversary generation. Therefore the sparsity of the resulting convex combination can be guaranteed (illustrated in Figure 4). We will add this discussion to our revision.
>
> Regarding $\alpha$ in our work, it controls the weight of sparsity regularization (the ablation study of which can be seen in Section 4.3). In particular, a larger $\alpha$ gives more emphasis on the sparsity during adversary generation, and thus aligns our modeled perturbation better with the discrete textual space. The better aligned modeling can further rule out some unnecessarily hard cases and thus we deem that it benefits both vanilla and robust accuracy (which is aligned with our result in Table 2). If $\alpha$ goes too large, the process of adversary generation will focus only on the sparsity, failing to find strong enough perturbations for robust training. In Zhou et al., the parameter you mentioned that contributes to higher robust accuracy but lower clean accuracy is $\lambda$. It controls the weight of 2nd hop between neighbors, which is not used in our method, and thus its effect may not be comparable with that of our $\alpha$.
>
> Regarding the access to substitution set, our setting is aligned with SOTA defense Jia et al. 2019 (Certified Robustness to Adversarial Word Substitutions), i.e. the model have the access to the set of all possible substitutions during the training. This is for fair comparison between defense techniques as it rules out the effect of using different substitution set. Besides, our proposed method can be applied to architectures like Transformers as long as its first layer is word embedding. We are running experiments on BERT and will update the results as a supplement.
>
> Regarding logit pairing, it improves the performance compared to normal adversarial training, but the margin is not significant. For example, on IMDB under Genetic attacks based on LSTM, using logit pairing ($\beta=4$) for our method achieves 79.0% robust accuracy, while using normal adversarial training achieves 78.2%. All the performances are higher than previous SOTA.

---

### Author Response · Authors · 2020-11-22
**Summary of the updates to the revision**

Hi all,

We have updated our manuscript, and the changes are summarized as follows:
 (1) We have added new experiments on BERT in Appendix B.2. (2) We have evaluated the performance under Genetic attacks without language model constraint, which is reported in Appendix B.1. (3) We have evaluated the reduced perturbation region by convex hull compared to $l_2$ ball and hyper-rectangle, and the analysis and results are reported in Appendix B.3. (4) We have updated the related work in Section 5.

We thank all the reviewers again for their insightful review and valuable feedback.

The authors.

---

### Decision · Program_Chairs · 2021-01-07
**Final Decision**

**Decision:**

Accept (Spotlight)

**Comment:**

All three reviewers are positive, and the authors have addressed essentially all the questions raised by the reviewers. The main insight of the paper is clear, and the empirical results are good, so a spotlight is deserved.